# Absorption and Biotransformation of Selenomethionine and Selenomethionine-Oxide by Wheat Seedlings (*Triticum aestivum* L.)

**DOI:** 10.3390/plants13030380

**Published:** 2024-01-27

**Authors:** Qi Wang, Siyu Huang, Qingqing Huang, Yao Yu, Huafen Li, Yanan Wan

**Affiliations:** 1Beijing Key Laboratory of Farmland Soil Pollution Prevention and Remediation, College of Resources and Environmental Sciences, China Agricultural University, Beijing 100193, China; wangqi88@cau.edu.cn (Q.W.); huangsiyu@cau.edu.cn (S.H.); lihuafen@cau.edu.cn (H.L.); 2Innovation Team of Heavy Metal Ecotoxicity and Pollution Remediation, Ministry of Agriculture and Rural Affairs (MARA), Agro-Environmental Protection Institute, MARA, Tianjin 300191, China; huangqingqing@caas.cn; 3School of Resources and Environmental Engineering, Hefei University of Technology, Hefei 230009, China; valcoyu@126.com

**Keywords:** wheat, selenomethionine, selenomethionine-oxide, uptake, Se speciation, interaction

## Abstract

An in-depth understanding of Se uptake and metabolism in plants is necessary for developing Se biofortification strategies. Thus, hydroponic experiments were conducted to investigate the associated processes and mechanisms of organic Se (selenomethionine (SeMet) and selenomethionine-oxide (SeOMet)) uptake, translocation, transformation and their interaction in wheat, in comparison to inorganic Se. The results showed that Se uptake by the roots and the root-to-shoot translocation factor under the SeMet treatment were higher than those under the selenite, selenate and SeOMet treatments. The uptake and translocation of SeMet were higher than those of SeOMet within 72 h, although the differences gradually narrowed with time. The uptake of SeMet and SeOMet was also sensitive to the aquaporin inhibitor: AgNO_3_ addition resulted in 99.5% and 99.9% inhibitions of Se in the root in the SeMet and SeOMet treatments, respectively. Once absorbed by the root, they rapidly assimilated to other Se forms, and SeMet and Se-methyl-selenocysteine (MeSeCys) were the dominant species in SeMet- and SeOMet-treated plants, while notably, an unidentified Se form was also found in the root and xylem sap under the SeMet treatment. In addition, within 16 h, SeOMet inhibited the uptake and translocation of SeMet, while the inhibition was weakened with longer treatment time. Taken together, the present study provides new insights for the uptake and transformation processes of organic Se within plants.

## 1. Introduction

Selenium (Se) is an essential micronutrient for human and animal health [1], associated with multiple physiological functions such as antioxidation, anti-carcinogenesis and immunoregulation, while a deficiency in Se can lead to the risk of cancer and cardiovascular diseases [2,3]. The World Health Organization recommends a daily Se intake of 50–200 μg d^−1^ for adults, as the optimum level of Se in living organisms is quite narrow in range. However, due to the significant variances in soil Se contents worldwide and the distribution of Se-poor soils in some populous regions, Se deficiency has become a public health problem, and approximately one billion people suffer from insufficient Se intake [4,5]. Food fortification strategies should be developed and recommended to increase Se content in the human diet, especially in low-Se areas. Agronomic biofortification with Se-containing fertilizers supplemented in soil or substrates is the most effective way to increase Se levels, and it has also been widely applied [6,7,8].

Se accumulation in plants mainly depends on the amount and availability of Se in soil, which are associated with the different chemical forms of Se, and the plant species [9]. Se naturally occurs as selenide, elemental Se, selenite, selenate and organic Se, and their distributions are governed by soil conditions, including pH, Eh, Fe/Mn/Al oxides, adsorbing surfaces, organic matter and microbial activity [10]. The uptake and biotransformation mechanisms of selenate and selenite have been well established. Selenate is transported through the cell membrane into the roots via sulfate transporters [11]. And it continues to be transported in the form of Se (VI) via the xylem to the shoots [12]. Nevertheless, selenite is taken up by the root via phosphate and silicon transporters [13,14]. Then, it can easily be converted into organic Se and rarely transported to the shoots [15,16]. Organic forms of Se, such as selenomethionine (SeMet) and selenomethionine Se-oxide (SeOMet), also make up an important fraction of available Se in soil, and they are readily absorbed by the plant root [9,17,18]. However, the uptake and biotransformation mechanisms of organic Se in plants remain to be experimentally confirmed.

Previous reports consistently revealed that the uptake efficiency of SeMet was much higher than that of selenate, selenite and SeOMet [19,20,21]. Our latest result revealed that SeMet uptake was markedly inhibited by the aquaporin inhibitor (AgNO_3_), indicating that rice absorbs SeMet primarily via aquaporin. In comparison, SeOMet, a derivative obtained from the transformation of SeMet, is taken up by an energy-dependent symport process and imported by rice roots via aquaporin and the K^+^ channel. When simultaneously supplied with both SeOMet and SeMet, SeOMet appeared to inhibit the uptake and translocation of SeMet [22]. Moreover, a recent report revealed that NRT1.1B was involved in SeMet translocation from the root to shoot [23]. Taken together, the above research on organic Se primarily concentrates on rice, but the mechanism and application of organic Se in wheat are poorly understood.

In contrast with rice, wheat is an aerobic plant with a significantly different structure of the stem and root. Meanwhile, wheat can produce more SeMet by absorbing Se in the soil or foliage and storing them in grains, thus providing the major source of Se in dietary intake [24,25]. In this study, the assimilation of SeMet and SeOMet by wheat in terms of uptake, translocation and biotransformation was reported through a series of hydroponic experiments, and the main purpose of this study was to investigate (1) the difference between the uptake and translocation of organic Se (SeMet and SeOMet) and inorganic Se (selenite and selenate) in wheat plants, (2) the distribution and speciation of Se in plants treated with organic Se, and (3) the interaction between SeMet and SeOMet during the uptake and transformation processes, as well as provide a scientific basis for practical Se biofortification with some help in the application and development of Se resources.

## 2. Materials and Methods

### 2.1. Plant Culture and Experimental Conditions

Seeds of wheat (*Triticum aesticum* L. cv. Luyuan502, a semi-winter middle-late cultivar) were surface-sterilized in 10% (*v*:*v*) NaClO solution for 15 min, rinsed thoroughly with deionized water and impregnated into saturated CaSO_4_ solution overnight, and then germinated on the pre-sterilized floating PVC screen in a solution of 0.5 mmol L^−1^ CaCl_2_. Seven days after germination, four wheat seedlings were transplanted to a plastic pot containing 3 L 1/5 Hoagland nutrient solution. Its nutrient concentration was as follows (mmol L^−1^): 1.0 KNO_3_, 0.1 KH_2_PO_4_, 0.457 MgSO_4_·7H_2_O, 1.0 Ca(NO_3_)_2_·4H_2_O, 1 × 10^−3^ ZnSO_4_·7H_2_O, 0.060 Fe(III)-EDTA, 3 × 10^−3^ H_3_BO_3_, 0.2 × 10^−3^ CuSO_4_·5H_2_O, 1 × 10^−3^ (NH_4_)_6_Mo_7_O_24_·4H_2_O, 1 × 10^−3^ MnSO_4_·H_2_O. The pH of this solution was adjusted to 6.0 with 2 mmol L^−1^ MES (2-morpholinoethanesulphonic acid, pH was adjusted with 1 mmol L^−1^ KOH or 1 mmol L^−1^ HCl). The solution was aerated continuously and renewed every three days, throughout the experiment. Plants were grown in a greenhouse with controlled conditions as follows: 20 °C/15 °C day/night temperatures, respectively; 14 h d^−1^ photoperiod with a light intensity of 240–350 μmol m^−2^ s^−1^; and 60% of relative humidity. Seeds of wheat were provided by Shandong Academy of Agricultural Sciences, China.

### 2.2. Comparison of Uptake and Translocation between Organic Se and Inorganic Se in Wheat Weedlings

Four-week-old wheat seedlings were transferred to pots (one plant per pot) containing 1 L of aerated uptake solutions, which were composed of normal nutrients and different species of Se (5 μmol), including selenite, selenate, SeMet and SeOMet. Each treatment had three replicates. The pH of these uptake solutions was buffered to 6.0 with 2 mmol MES. After treatment for 24 h, the seedling roots were rinsed with deionized water three times and then transferred to 150 mL ice-cold desorption solutions (1 mmol L^−1^ CaSO_4_ + 1 mmol L^−1^ K_2_HPO_4_, 2 mmol L^−1^ MES, pH 6.0) for 15 min to remove the Se and other ions adsorbed on the root surface. After desorption, the wheat plants were rinsed thoroughly with deionized water and then separated into shoots and roots, weighed, frozen in liquid nitrogen, powdered and analyzed for total Se contents. Selenite, selenate and Selenomethionine (SeMet) were obtained from Sigma-Aldrich (St Louis, MO, USA). Selenomethionine-oxide (SeOMet) was prepared according to Larsen et al. (2004) [26].

### 2.3. Effects of Aquaporin Inhibition on Uptake of SeMet and SeOMet

The aquaporin inhibition assay was performed according to the reported method with slight modification [27]. Four-week-old seedlings were transferred to absorption solutions containing 5 μmol L^−1^ organic Se (SeMet or SeOMet) and 0 or 100 μmol L^−1^AgNO_3_ (2 mmol L^−1^ MES, pH 6.0), and thus, there were four treatments in total: SeMet alone, SeMet + AgNO_3_, SeOMet alone and SeOMet + AgNO_3_. Three replicates were used for each treatment. After 24 h, wheat roots were rinsed with deionized water and the Se adsorbed on the root surface was desorbed as described earlier. Then, shoots and roots were separated and their total Se contents were analyzed.

### 2.4. Uptake, Transformation and Interaction of SeMet and SeOMet in Wheat Seedlings at Different Times

This experiment was conducted to investigate the transformation regulation of Se speciation, and whether SeMet and SeOMet interact during uptake and assimilation by wheat plants. Four-week-old wheat seedlings were exposed to 1 L pots (one plant per pot) containing aerated uptake solutions for 4, 16, 32 and 72 h. The uptake solutions were composed of normal nutrients and 10 μmol L^−1^ SeMet, 10 μmol L^−1^ SeOMet or 10 μmol L^−1^ SeMet + 10 μmol L^−1^ SeOMet (2 mmol L^−1^ MES, pH 6.0). Each treatment was replicated in three pots. After the organic Se absorption, the roots were rinsed with deionized water and desorbed as described above. The fresh shoots and roots were weighed, frozen in liquid nitrogen, powdered and stored at −80 °C for subsequent determination of total Se and Se speciation. In addition, when treated for 16 h, wheat shoots were cut at 2 cm above the roots, and the xylem saps were collected using a 1 mL pipette for 2 h. After collection, the xylem saps were stored at −80 °C for the analysis of Se speciation.

### 2.5. Analysis of Total Se and Se Speciation in Plant Tissues

For the analysis of total Se content, powdered plant samples were digested with HNO_3_ (GR) using microwave sample preparation system (CEM, MARS5, CEM Corp., Matthews, NC, USA). Total Se content in wheat tissues was determined based on the method of Fujii et al. (1988) [28]. During the whole analysis process, a certified reference material (GBW10014) and blanks were included for quality assurance. The recovery for GBW10014 was 85–110%.

Powdered fresh plant samples (0.4000 g) were extracted with 5 mL of 8 mg mL^−1^ protease XIV (Sigma-Aldrich, USA) in an oscillation box (37 °C, 125 rpm) for 24 h. The extracts were subsequently centrifuged at 12,000 rpm for 15 min and then filtered through 0.22 μm filters (Millipore, Billerica, MA, USA). The xylem saps were also filtered through 0.22 μm filters. Selenium speciation in plant extracts and xylem sap was determined using HPLC-ICP-MS (Agilent LC1260 series and Agilent ICP-MS 7700, Agilent Technologies, Santa Clara, CA, USA).

Selenium species were separated using an anion exchange chromatography column (Hamilton PRP-X100, Hamilton Company, Reno, NV, USA), and the outlet of the separation column was connected to an ICP-MS detection system. The mobile phase was 40 mmol L^−1^(NH_4_)_2_HPO_4_ (pH 6.0). The Se standards, namely, Se (IV) (Na_2_SeO_3_), Se (VI) (Na_2_SeO_4_), SeMet (selenomethionine), SeCys_2_ (selenocysteine) and MeSeCys (Se-methyl-selenocysteine), were obtained from the National Research Center for Certified Reference Materials, Beijing, China. SeOMet (Selenomethionine Se-oxide) was prepared as per Larsen et al. [26]. The chemical species of Se in samples were identified and quantified by retention times and peak areas, respectively.

### 2.6. Data Analysis

Total Se concentration (*T_Shoot_*, *T_Root_* and *T_Se_*) and the transfer factor (*TF*) of Se were calculated using Equations (1)–(4):*T_Shoot−Se_* = *C_Shoot−Se_* × *Biomass_Shoot_*
(1)
*T_Root−Se_* = *C_Root−Se_* × *Biomass_Root_*(2)
*T_Se_* = *T_Shoot−Se_* + *T_Root−Se_*(3)
*TF* = *C_Shoot−Se_*/*C_Root−Se_*(4)
where *C_Shoot_*_−_*_Se_* and *C_Root−Se_* represent the Se content in wheat shoot and root, respectively.

The proportion of Se distribution in wheat tissues was calculated using Equations (5) and (6):*Shoot* − *Se*% = *T_Shoot−Se_*/*T_Se_* × 100%(5)
*Root* − *Se*% = *T_Root−Se_*/*T_Se_* × 100%(6)

The proportion of each Se species (*Pro_i_*) in wheat was calculated using Equation (7):*Pro_i_* = *C_i_*/*C_Shoot/Root−Se_* × 100% (7)
where *C_i_* and *C_Shoot/Root_*_−_*_Se_* represent the content of a certain Se species and the sum of the contents of the four Se species in the wheat tissues, respectively.

All data are presented as means ± standard errors (SEs; *n* = 3). Analysis of variance was performed using SAS 9.3 (least significant difference, *p* < 0.05).

## 3. Results

### 3.1. Uptake and Translocation of Different Se Treatments in Wheat Seedlings

To evaluate the uptake and translocation of organic Se versus inorganic Se, Se (IV), Se (VI), SeMet and SeOMet were supplied in this experiment. The results showed that the Se contents in the wheat shoots and roots were significantly affected by Se forms, and the level of Se in the root was obviously higher than that in the shoot (Figure 1). In the wheat root, the Se content in the SeMet treatment was 65.4%, 69.0% and 7.09 times higher than that in the SeOMet, Se (IV) and Se (VI) treatments, respectively, while the differences in the shoot were even higher (10.2, 37.5 and 28.1 times).

The translocation and distribution of Se in wheat plants were also significantly affected by Se forms (Table 1). In the Se (IV) treatment, most of the Se was distributed to the root, accounting for 94.0% of the total Se uptake by wheat seedlings. In contrast, more the half of the Se was distributed to the shoot (57.6%) in the SeMet treatment. No significant difference was obtained between the Se (VI) and SeOMet treatments, and the former was slightly higher in the Se distribution in the shoot. The translocation factor from the root to shoot in the SeMet treatment was 2.61, 4.22 and 22.0 times higher that of the Se (VI), SeOMet and Se (IV) treatments, respectively.

### 3.2. Effects of Aquaporin Inhibition on Uptake of SeMet and SeOMet

To examine the uptake of SeMet and SeOMet by wheat through water channels, 0.1 mmol L^−1^ AgNO_3_ was used as a water channel blocker [29]. The Se contents in wheat plants were measured after 24 h of exposure in aquaporin inhibition and two organic Se forms (Figure 2). Generally, the Se contents in the wheat root and shoot in the SeMet treatment were higher than those in the SeOMet treatment, and those of the former were 20.4% and 1.63 times higher those of the latter, respectively, in the Se-alone treatment. Compared with the control (Se-alone treatment), AgNO_3_ addition dramatically reduced the Se contents in the root and shoot by 99.5% and 96.4% in the SeMet treatment, respectively, while the differences in the SeOMet treatment were 99.9% and 92.2%, respectively.

### 3.3. Uptake, Translocation and Interaction of SeMet and SeOMet in Wheat Seedlings in Different Time

The Se contents in wheat plants were significantly affected by Se forms, exposure time and the interactions between these two factors (Figure 3, *p* < 0.05). Irrespective of the Se treatments, the root and shoot Se increased gradually with increasing exposure time. When the treatment time was ≤32 h, the Se in the wheat root treated with SeMet was significantly higher than that treated with SeOMet, while the difference decreased with time, from 81.3% to 22.5% (*p* < 0.05); when treated for 72 h, there was no significant difference between the two Se forms (Figure 3a). Similarly, the shoot Se content in the SeMet treatment was higher than that in the SeOMet treatment at all Se-treated time points, while the difference gap narrowed with time (Figure 3b). When treated for 4 h, only a small proportion of Se was transferred to the shoot; with prolonged exposure time, the proportions of Se distributed to the shoot were 18.1–56.8%. In addition, the transfer factor and shoot-Se% in the SeMet treatment were higher than those in the SeOMet treatment at all exposure times (Figure 4).

In the experiment, both SeMet and SeOMet were applied to the nutrient solution to investigate the interaction between SeMet and SeOMet in Se uptake and translocation processes. When the exposure time was ≤16 h, the root and shoot Se in the SeMet/SeOMet treatment were lower than those in the single-SeMet treatment, by 3.0–45.2% and 31.0–95.5%, respectively, while they were higher than those when treated over 32 h (except for the root at 72 h). No significant differences of Se in plants were found between SeMet/SeOMet and single-SeOMet treatments at 4 h, and after 16 h, the former was higher than the latter (except for the root at 72 h) (Figure 3). The transfer factor and shoot-Se% of plants treated with SeMet/SeOMet were lower than those in the single-SeMet treatment when the exposure time was ≤32 h, while they were higher than those at 72 h. And the transfer factor and shoot-Se% of the mixed Se treatment were higher than those of the single-SeOMet treatment at 32 h and 72 h (Figure 4).

### 3.4. Transformation of SeMet and SeOMet in Wheat Seedlings

Under the current working conditions, MeSeCys, Se (IV), SeMet and Se (VI) were ideally separated through HPLC-ICP-MS, and the retention times were 186 s (RT_186_), 274 s (RT_274_), 374 s (RT_374_) and 756 s (RT_756_), respectively. However, the retention times for SeOMet and SeCys_2_ were close at 141 s (RT_141_) and 144 s (RT_144_), respectively. Therefore, the standard curve for SeOMet was tested and plotted solely (Appendix A). In addition, an unidentified Se compound (the first peak on the chromatogram, RT_121_) was found in plants treated with SeMet and SeMet/SeOMet, especially in the wheat root (Figure 5).

Overall, the proportions of SeMet, MeSeCys, Se (IV) and SeCys_2_ or SeOMet in the total selenium content in wheat roots were 10.3–25.8%, 3.65–21.7%, 0.10–0.66% and 0.19– 1.44%, respectively. Similarly, the corresponding proportions in the shoot were 31.1–87.8%, 5.22–11.4%, 0.86–1.75% and 0.55–2.19%. In the wheat roots, the contents of MeSeCys and SeMet in the single-SeMet treatment were 66.7% and 89.1% higher than those in the single-SeOMet treatment, respectively, while the Se (IV) content in the former was lower than that of the latter, although not significantly. In addition, the contents of MeSeCys, Se (IV) and SeCys_2_ or SeOMet in the SeMet/SeOMet treatment were higher than those in the single-SeMet or single-SeOMet treatment: MeSeCys in the SeMet/SeOMet treatment was 2.27 and 4.45 times higher than those in single-SeMet and single-SeOMet treatments, respectively (*p* < 0.05), while the SeMet content in the SeMet/SeOMet treatment was similar to that in the single-SeMet treatment. In the wheat shoot, the content of SeMet in the single-SeMet treatment was 2.06 times higher than that in the single-SeOMet treatment (*p* < 0.05), while the other Se forms (MeSeCys, Se (IV), SeCys_2_ or SeOMet) in the former were lower than those in the latter. Under the SeMet/SeOMet treatment, the Se forms of MeSeCys and Se (IV) were higher than those in the single-SeMet or single-SeOMet treatment, while SeMet and SeOMet or SeCys_2_ were similar to those in the single-SeMet treatment (Table 2).

### 3.5. Se Concentration and Speciation in Xylem Saps

After 16 h of exposure, Se speciation in the xylem sap was determined, and the unidentified Se compound (the first peak on the chromatogram, RT_121_) was also found therein (Figure 6). In the SeMet-alone treatment, MeSeCys, Se (IV), SeMet and the unknown Se form were detected, and the concentrations were as follows: unknown Se form > MeSeCys > Se (IV) > SeMet. When SeOMet was added alone, however, only an unknown Se form was detected in xylem sap, whose concentration was much lower than that of the SeMet treatment. Under the SeMet/SeOMet treatment, the Se forms in xylem sap were same as those under the SeMet treatment, while the concentration of MeSeCys in the former was 90.3% lower than that in the latter (Table 3).

## 4. Discussion

### 4.1. Difference in The Uptake and Translocation between Organic Se and Inorganic Se in Wheat Seedlings

In this study, the uptake and translocation characteristics of organic Se (SeMet and SeOMet) by wheat plants were varied compared to those of inorganic Se (selenite and selenate). The order of the SeMet uptake ability by the wheat root was much higher than those of SeOMet, selenite and selenate (Figure 1 and Table 1). Huang et al. (2015) observed that rice roots could absorb more selenite than selenate [30]. Some reports have indicated that the uptake of SeMet by maize, wheat and rice roots was much faster than that of selenite or selenate [31,32,33]. The differences in the uptake could be attributed to differences in the activities of their respective transporters, none of which, however, are Se-specific. Selenate is taken up by the roots via sulfate transporters [11], and selenite is absorbed in an active process mediated by phosphate and silicon transporters [13,14,15]. On the other hand, SeMet, as a seleno amino acid, might be absorbed via root amino acid transporters [24,34]. Our study showed that the uptake of SeMet and SeOMet was sensitive to the aquaporin inhibitor, such as AgNO_3_ that is reported to partially inhibit the uptake of selenite and Se nanoparticles in crops [27,35,36]. In this study, the AgNO_3_ addition resulted in 99.5% and 99.9% of inhibition of Se in the root in SeMet and SeOMet treatments, respectively (Figure 2), indicating that the influx of SeMet and SeOMet might be mediated via aquaporins. The phenomenon that the plant root could absorb more organic Se than inorganic Se might be attributed to the following mechanisms: (1) transporter-mediated SeMet and SeOMet intakes are more active than those of inorganic Se [19]; (2) the uptake of inorganic forms might be inhibited by sulfate and phosphate in the culture solution [15,37]; and (3) inorganic forms such as selenate and selenite are more toxic to plants than SeMet and SeOMet, resulting in less intake due to the defensive responses of plants [32,38].

In rice seedlings exposed to different sources of Se for 24 h, we found that the proportions of Se distributed in the wheat shoot (and transfer factor from root to shoot) decreased in the order SeMet > selenate > SeOMet > selenite (Table 1). Sulfate transporters (such as Sultr2;1, Sultr3;5 and Sultr1;3) and phosphorus transporters (such as OsPT8) are the main transporters involved in the translocation of selenate and selenite from the root to shoot, respectively [11,39,40]. In addition, Zhang et al. (2019) found that NRT1.1B, which is associated with nitrate uptake and transport, mediates the transport activity of SeMet in rice [23]. Upon uptake, most selenite is quickly metabolized into organic Se compounds and retained in the roots, whereas selenate can be translocated to the shoots rapidly [15,41,42]; thus, the transport capacity of Se in plants treated with selenate was greater than that treated with selenite [30,43,44]. In the study of Xu et al. (2020), the translocation of Se from the rice root to shoot in the SeMet treatment was higher than that in the selenate or selenite treatment [33], and Kowalska et al. (2020) reported that the TF of Se in lettuce treated with SeMet was 3.56 times higher than that treated with selenite, which are consistent with our study [20]. However, in another study, Se in the shoot originating from SeMet was higher than that originating from selenate in garlic (*Allium sativum*), while it tended to be lower than that originating from selenate in Indian mustard (*Brassica juncea*) [45]. Wang et al. (2020) reported that the order of Se-shoot% in maize seedling was selenate treatment > selenite treatment > SeMet treatment, and selenate treatment > SeMet treatment > selenite treatment when 0.01 mg L^−1^ and 0.1 mg L^−1^ Se are supplied, respectively [32]. Therefore, the transport capacity of plants not only depends on Se forms but also on plant species and Se concentrations.

As for the two organic Se forms, the uptake and transport of SeMet were greater than those of SeOMet within 72 h (Figure 1, Figure 2 and Figure 3), while the difference gradually narrowed with time (Figure 3). This is probably because the uptake of SeMet gradually approaches saturation during the treatment time, and the uptake rate decreases. In addition, the effect of exposure time on organic Se absorption might be attributed to the transformation between SeOMet and SeMet in the plant root or rhizosphere. It has been demonstrated that SeOMet was detected in the root of lettuce (*Lactuca sativa* L.) exposed to SeMet [20]. And in this study, SeMet was detected in the wheat root in SeOMet, while SeOMet or SeCys_2_ was also detected in the root treated by SeMet (Figure 5), thereby indicating the occurrence of the oxidative and reductive transformation of Se in plants.

### 4.2. Transformation of SeMet and SeOMet in Wheat Seedlings

In the present study, MeSeCys, Se (IV), SeMet and Se (VI) were quantitatively and qualitatively measured by HPLC-ICP-MS, while SeCys_2_ and SeOMet cannot be identified clearly, as their retention times overlapped (Appendix A), which was also found in previous studies [46,47]. Although it was difficult to identify, the chromatograms showed that the contents of these two Se forms were low in wheat plants (Figure 5 and Figure 6). In addition, there was a difference between the sum of the identified peaks and the total Se in the plant (Table 2), which might be due to the limitation of standard Se compounds (unknown compound) and the low extraction efficiency of protease XIV.

Plants accumulate Se in different chemical forms, and the speciation of Se in plants depends on the plant species and the Se forms in their surroundings [48]. It has been proved that plants can absorb inorganic Se (i.e., selenate and selenite) and nano-Se and convert to organic Se [26,35,49]; for example, SeMet, MeSeCys and SeCys_2_ were all detected in the wheat root and shoot treated with selenite [35]. As for organic Se, SeMet and MeSeCys were the dominant forms in wheat in the SeMet or SeOMet treatment in our study (Figure 5 and Table 2). SeMet and MeSeCys are well assimilated by humans and animals and are advocated for use as a nutritional selenium supplement [50]. Furthermore, it has been reported that MeSeCys has anticarcinogenic and antitumor activities [51], and the results showed that the pharmaceutically valuable MeSeCys can be efficiently biosynthesized in plants fortified with SeMet (or SeOMet).

During Se assimilation in plants, a variety of intermediates are produced by organisms, such as selenohomocysteine (SeHCys), selenocystathionine (SeCysTH) and Se-adenosyl-selenomethionine (SeAM) [9,52]. As shown in Figure 5 and Figure 6, an obvious peak (RT_121_) corresponding to an unknown Se compound was detected in the wheat root and xylem sap in the SeMet or SeMet/SeOMet treatment, but not in the SeOMet treatment (except for xylem sap). Therefore, we speculated that SeMet transformed into an intermediate after being absorbed by the root, and the intermediate could be both easily transformed into other forms (such as MeSeCys and Se (IV)) in the root and transfer to the shoot, and then be converted to other Se forms in the shoot. As for SeOMet, however, once absorbed by the root, it is rapidly converted to SeMet, which becomes the main rate-limiting step of Se assimilation. Due to this technical condition restriction, the possible molecular formula of this unknown species needs to be determined using a higher-sensitivity instrument in the future, such as UHPLC-ESI-Orbitrap MS [19].

### 4.3. Interaction between SeMet and SeOMet in Wheat Seedlings

The results showed that Se contents in the wheat root and shoot treated with SeMet/SeOMet were lower than those in the single SeMet treatment (within 16 h) (Figure 3 and Figure 4), and at 16 h, the MeSeCys content in the SeMet/SeOMet treatment was lower than that in the SeMet treatment (Figure 6), indicating that an interaction between SeMet and SeOMet occurred. Previous studies have reported non-additive effects in the uptake and translocation of different Se forms; for example, a certain interaction was exhibited between selenate and selenite. In previous studies, selenite inhibited the uptake of selenate by wheat, when both Se forms were supplied [15,43]; a consistent phenomenon was also observed in tomato (*S. lycopersicum* L.) [12]. Similarly, in the present study, we found that the presence of SeOMet appeared to suppress the accumulation of SeMet in wheat. The interactions between different Se forms might be caused by the specific absorption of plants: an optimal absorption scheme is adopted based on intrinsic synergistic action in response to mixed supplements of different Se forms, thus conserving the enzymes and energy required for subsequent Se assimilation [53].

In addition, the interactions between SeMet and SeOMet in wheat depended on the exposure time. Within 16 h of treatment, SeOMet inhibited the uptake and translocation of SeMet, while, as the treatment time was prolonged (32 and 72 h), the inhibition was weakened (Figure 3, Figure 4, Figure 5 and Figure 6). This might be explained by the metabolism of Se in plants: in a short time (≤16 h), the saturation threshold of the corresponding enzyme was not reached, so the rate of SeMet uptake was high; in this case, the SeOMet addition would have a negative effect on root SeMet absorption. After 32 h, the SeMet uptake rate gradually declined, and the transformation between SeMet and SeOMet in the plant root or rhizosphere might also account for the decrease in interaction. Furthermore, at 32 h, the Se content in the root treated with SeMet/SeOMet was significantly higher than that of the SeMet treatment, while at 72 h, Se in the shoot of the former was higher than that of the latter (Figure 3). As we know, Se has a dual effect on plants, and the excessive accumulation of Se would disrupt the structure and function and cause cytotoxicity; thus, organisms might alleviate this damage by accumulating more Se in the root and regulating Se speciation [54]. Structural Se-containing amino acids (such as SeMet and SeCys) will rapidly convert to non-structural amino acids (such as MeSeCys), thereby improving plant tolerance [55]. The higher content of MeSeCys in the SeMet/SeOMet treatment also proved this, to a certain degree (Figure 5). However, this needs to be verified in further studies.

## 5. Conclusions

The present study has revealed differences between SeMet and SeOMet with respect to Se uptake, translocation and speciation in wheat seedlings. Compared with SeOMet and inorganic Se forms (selenite and selenate), SeMet exhibited a greater uptake and transport capacity, though the differences between the two organic Se forms (SeMet and SeOMet) narrowed with time. And the influx of SeMet and SeOMet might be mediated via aquaporins. Speciation analysis showed that SeMet and MeSeCys were the dominant species in all wheat plants, while an unidentified Se species was also found in the root and xylem sap of plants treated with SeMet. In addition, we also found that when simultaneously supplied with both SeOMet and SeMet, SeOMet appeared to inhibit the uptake and translocation of SeMet, while the interaction was weakened with time. We anticipate that the findings of this study would provide a theoretical basis for developing efficient Se-enriched foods and resolving Se-deficiency in humans and animals.

## Figures and Tables

**Figure 1 plants-13-00380-f001:**
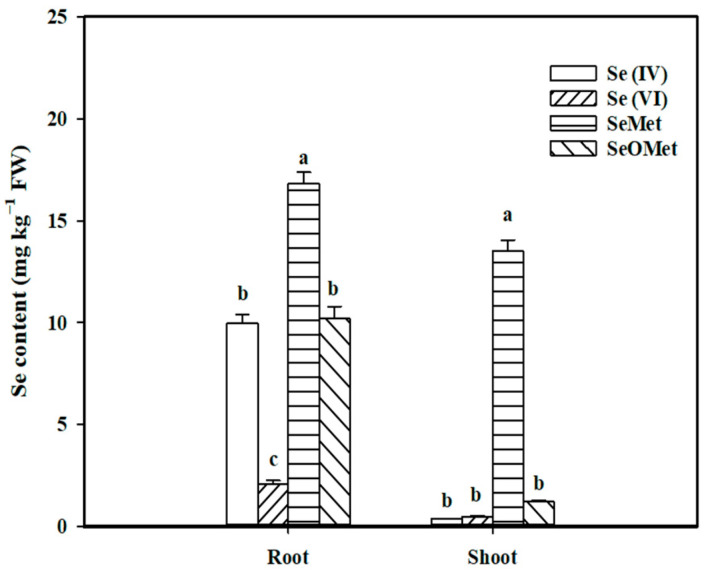
The Se content in root and shoot of wheat seedlings under different Se treatments. Data are means + SE (*n* = 3). The different lowercase letters indicate significant differences among Se treatments at *p* < 0.05. Se (IV), selenite; Se (VI), selenate; SeMet, selenomethionine; SeOMet, selenomethionine-oxide.

**Figure 2 plants-13-00380-f002:**
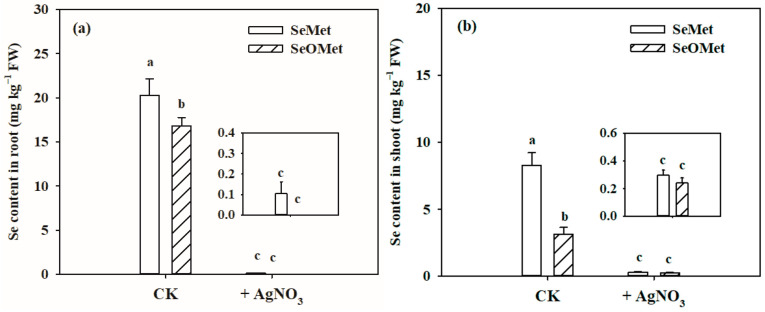
Effects of aquaporin inhibition of AgNO_3_ on Se contents in the wheat root (**a**) and shoot (**b**) under SeMet and SeOMet treatments. Data are means + SE (*n* = 3). The different letters indicate significant differences among the Se treatments at *p* < 0.05. SeMet, selenomethionine; SeOMet, selenomethionine-oxide.

**Figure 3 plants-13-00380-f003:**
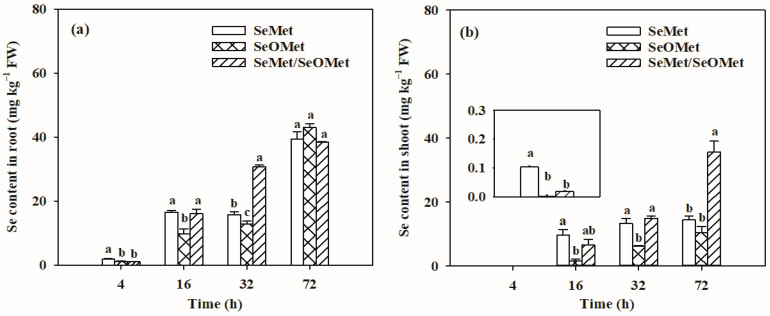
Se contents in root (**a**) and shoot (**b**) of wheat seedlings treated with different times and organic Se. Data are means + SE (*n* = 3). The different letters in individual treatment times indicate significant differences among the organic Se treatments at *p* < 0.05. SeMet, selenomethionine; SeOMet, selenomethionine-oxide.

**Figure 4 plants-13-00380-f004:**
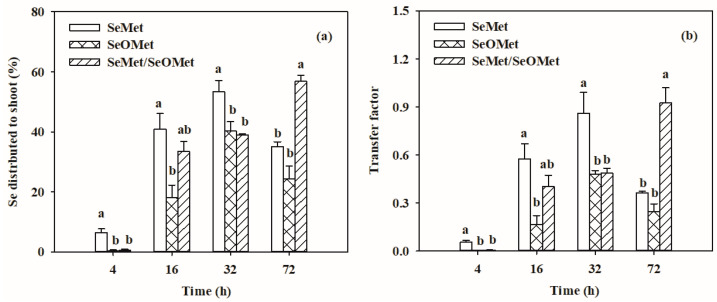
Proportion of Se distributed to shoot (**a**) and transfer factor of Se (**b**) in wheat seedlings treated with different times and organic Se. Data are means + SE (*n* = 3). The different letters in individual treatment times indicate significant differences among the organic Se treatments at *p* < 0.05. SeMet, selenomethionine; SeOMet, selenomethionine-oxide.

**Figure 5 plants-13-00380-f005:**
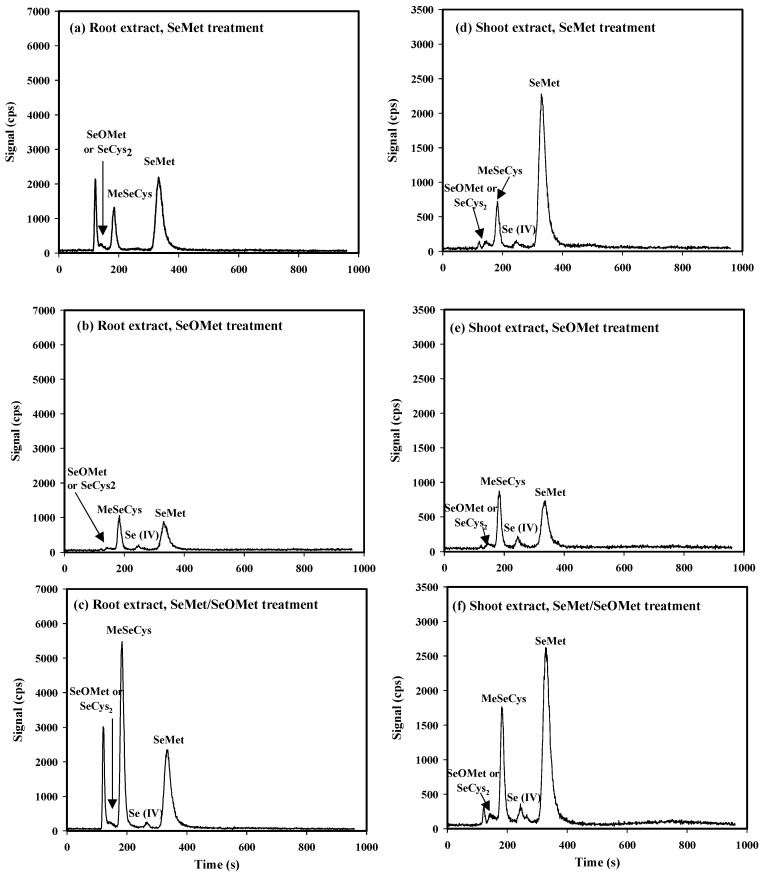
Examples of HPLC-ICP-MS chromatograms of Se speciation in wheat roots and shoots treated for 72 h.

**Figure 6 plants-13-00380-f006:**
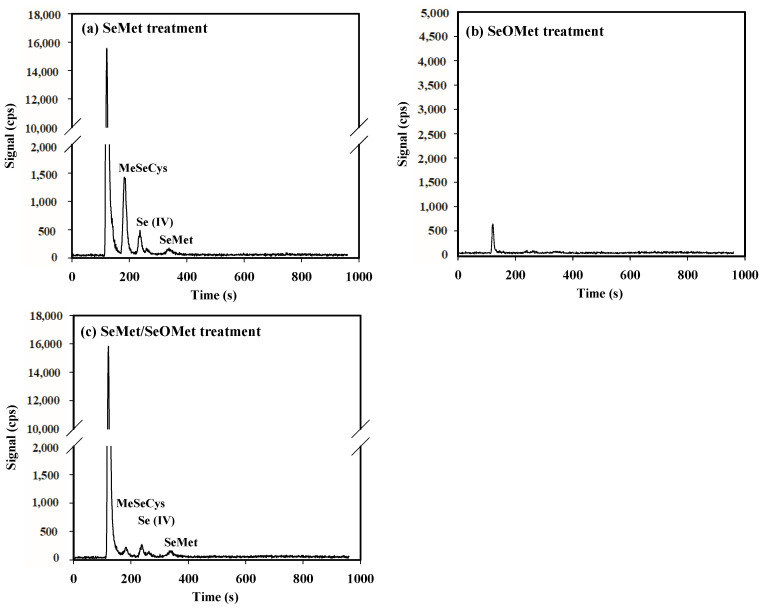
Examples of HPLC-ICP-MS chromatograms of Se speciation in the xylem sap from wheat seedlings treated for 16 h.

**Table 1 plants-13-00380-t001:** Effect of different Se forms on Se translocation and distribution in wheat seedlings.

Se Form	Se Distribution (%)		Transfer Factor
	Root	Shoot	
Se (IV)	94.0 ± 0.55 a	6.00 ± 0.63 c	0.028 ± 0.004 c
Se (VI)	71.3 ± 0.29 b	28.7 ± 0.34 b	0.178 ± 0.004 b
SeMet	42.4 ± 1.03 c	57.6 ± 1.19 a	0.643 ± 0.038 a
SeOMet	77.1 ± 0.83 b	22.9 ± 0.96 b	0.123 ± 0.008 b

Data are means ± SE (*n* = 3). The different letters in the same column indicate significant differences among the Se treatments at *p* < 0.05.

**Table 2 plants-13-00380-t002:** Effect of organic Se species supplied on Se speciation in the protease XIV extract from wheat root and shoot treated for 72 h.

Treatment		Se Species (mg kg^−1^ FW)			
		MeSeCys	Se (IV)	SeMet	SeOMet or SeCys_2_
Root	SeMet	2.55 ± 0.21 b (6.84%)	0.04 ± 0.01 a (0.10%)	8.19 ± 1.14 a (22.0%)	0.10 ± 0.03 a (0.19%)
	SeOMet	1.53 ± 0.16 b (3.65%)	0.10 ± 0.06 a (0.24%)	4.33 ± 0.09 a (10.3%)	0.13 ± 0.03 a (0.30%)
	SeMet + SeOMet	8.34 ± 1.82 a (21.7%)	0.25 ± 0.01 a (0.66%)	9.89 ± 0.46 a (25.8%)	0.56 ± 0.25 a (1.44%)
Shoot	SeMet	0.68 ± 0.54 a (5.22%)	0.11 ± 0.07 a (0.86%)	11.6 ± 1.58 a (87.8%)	0.29 ± 0.16 a (2.19%)
	SeOMet	1.41 ± 0.15 a (11.4%)	0.22 ± 0.18 a (1.75%)	3.79 ± 0.81 b (31.2%)	0.07 ± 0.03 a (0.55%)
	SeMet + SeOMet	2.65 ± 0.41 a (7.60%)	0.50 ± 0.07 a (1.43%)	10.6 ± 0.43 a (31.1%)	0.20 ± 0.01 a (0.58%)

Values in parentheses represent Se species percentage in wheat shoots or roots calculated using the expression: Se species content/total tissue content × 100%. Data are means ± SEs (*n* = 3). Different letters of the same Se species indicate significant differences among the organic Se treatments at *p* < 0.05.

**Table 3 plants-13-00380-t003:** Se speciation in the xylem sap from wheat seedlings treated for 16 h.

Treatment	Se Concentration in Xylem Sap (mg L^−1^)		
	MeSeCys	Se (IV)	SeMet
SeMet	1.29	0.27	0.16
SeOMet	0.00	0.00	0.00
SeMet/SeOMet	0.13	0.15	0.21

## Data Availability

Data are contained within the article and Appendix A.

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
