# Peer review of "Absorption and Biotransformation of Selenomethionine and Selenomethionine-Oxide by Wheat Seedlings (Triticum aestivum L.)"

_plants, 2024, doi:10.3390/plants13030380_

Round 1

Reviewer 1 Report

Comments and Suggestions for Authors

Key words

Line 32 Please remove „interaction”. In my opinion, this word is too broad and is loosely related to manuscript

Line 84 In chapter 2.1. Plant Culture and Experimental Condition showed that only a one-year greenhouse experiment was conducted. In biological tests with plants, a minimum of 2 years is needed to confirm the plants' response to fertilization. A one-year experiment in microvases is insufficient.

Line 95 The authors should explain how it was studied that the nutrient solution needs to be supplemented every three days. The authors should explain too why they used SeMet or SeOMet, i.e. organic forms of selenium, for fertilization, if in practice applying such forms to the soil does not make sense.

Line 114-122 The authors should explain for what purpose the research with AgNO3 was carried out?

Figure 1 The SeMet marking on the chart is illegible

The discussion of the results is a well-written chapter

Author Response

Dear Editors and Reviewers:

  Thank you for your letter and for the reviewers’ comments concerning our manuscript entitled "Absorption and biotransformation of selenomethionine and selenomethionine-oxide by wheat seedlings (Triticum aestivum L.)" (Manuscript ID: 2783807). Those comments are all valuable and very helpful for revising and improving our paper, as well as the important guiding significance to our researches. According with your advice, we have carefully checked and revised the relevant part in manuscript. All the changes made according to reviewers’ comments are highlighted in yellow in the manuscript. Here below is our description on revision.

Reviewers' Comments to Author:

  • Line 32 Please remove “interaction”. In my opinion, this word is too broad and is loosely related to manuscript.

Answer: Revised. We have used the word inhibition to instead interaction, ensuring readers to understand the result clearly.

  • Line 84 In chapter 2.1. Plant Culture and Experimental Condition showed that only a one-year greenhouse experiment was conducted. In biological tests with plants, a minimum of 2 years is needed to confirm the plants' response to fertilization. A one-year experiment in microvases is insufficient.

Answer: Agree. Besides the one-year greenhouse experiment of wheat conducted in present study, we also conducted hydroponic experiment of rice  in the latest study (Wang et al., 2022). No matter the species of plants, we demonstrated that the uptake and translocation of Se by plants were significantly influenced by both Se chemical forms and treatment time. Moreover, SeOMet appeared to inhibit the uptake and translocation of SeMet by wheat and rice. Our findings both provide important insights into the mechanisms underlying the absorption and biotransformation of SeMet and SeOMet within plants.

  • Line 95 The authors should explain how it was studied that the nutrient solution needs to be supplemented every three days. The authors should explain too why they used SeMet or SeOMet, i.e. organic forms of selenium, for fertilization, if in practice applying such forms to the soil does not make sense.

Answer: In our experience of hydroponics experiments for many years, when wheat seedlings are young, the nutrient solution renewed once every a week is enough for the growth of seedlings. About two weeks later, the nutrient solution renewed once every three days with gradually grow of seedlings.

In our previous pot experiment (Li et al., 2010), SeMet and SeOMet were both determined in soil pore water and rice tissues, indicating that SeMet and SeOMet were important fractions of available Se in soil and could be absorbed by plant root readily. But the uptake and bio-transformation mechanisms of SeMet and SeOMet in wheat remains to be experimentally confirmed. Therefore, the major aim of present study is to investigate the difference mechanisms about absorption and bio-transformation of SeMet and SeOMet by wheat seedlings.

  • Line 114-122 The authors should explain for what purpose the research with AgNO3was carried out?

Answer: In our previous and the present researches, AgNO3 was usually carried out as a water channel blocker to examine if the uptake of nano-Se, SeMet and SeOMet by plant through water channels (Hu et al., 2018; Wang et al., 2022).

  • Figure 1 The SeMet marking on the chart is illegible.

Answer: Revised. We have adjusted the title of Figure 1, 2, 3, 4, as revised in Lines 196-197, 222, 242-243 and 259-260 of the manuscript, highlighted in yellow, ensuring readers to understand the figure clearly.

  • The discussion of the results is a well-written chapter.

Answer: Thanks for your praise and appreciate.

Sincerely,

Yanan Wan, Ph. D.

Corresponding author for the manuscript

Reviewer 2 Report

Comments and Suggestions for Authors

Minor changes on the manuscript.

Author Response

Dear Editors and Reviewers:

  Thank you for your letter and for the reviewers’ comments concerning our manuscript entitled "Absorption and biotransformation of selenomethionine and selenomethionine-oxide by wheat seedlings (Triticum aestivum L.)" (Manuscript ID: 2783807). Those comments are all valuable and very helpful for revising and improving our paper, as well as the important guiding significance to our researches. According with your advice, we have carefully checked and revised the relevant part in manuscript. All the changes made according to reviewers’ comments are highlighted in yellow in the manuscript. Here below is our description on revision.

Reviewers' Comments to Author:

  • Line 73: What is the meaning of “high-security”?

Answer: Revised, as revised in Lines 75 of the manuscript, highlighted in yellow. , The meaning of high-security is that SeMet is the dominant organic Se form of cereals, which is more beneficial for human health.

  • Line 89: Delete “every”

Answer: Revised. 

  • Line 94-95:Is it a different adjustment?

Answer: Revised. Specific description of administrator was included in manuscript.

  • Line 128:Delete the word “uptake”.

Answer: Revised.

  • Line 303~304: Add year of publication and delete the “dosage”.

Answer: Revised.

  • Line 327, 333, 334: Add year of publication.

Answer: Revised. We have further checked the whole manuscript and added related year of publication.

  • Line 327: Delete the word “activities”.

Answer: Revised. 

  • Line 373-389: The paragraph should be shortened since is not supported sufficiently by the data.

Answer: Revised, as revised in Lines 389-401 of the manuscript, highlighted in yellow. This comment is valuable and very helpful for revising and improving our paper.

Sincerely,

Yanan Wan, Ph. D.

Corresponding author for the manuscript